# A Novel Jeilongvirus from Florida, USA, Has a Broad Host Cell Tropism Including Human and Non-Human Primate Cells

**DOI:** 10.3390/pathogens13100831

**Published:** 2024-09-26

**Authors:** Emily DeRuyter, Kuttichantran Subramaniam, Samantha M. Wisely, J. Glenn Morris, John A. Lednicky

**Affiliations:** 1Department of Environmental and Global Health, College of Public Health and Health Professions, University of Florida, Gainesville, FL 32610, USA; emilyderuyter@ufl.edu; 2Emerging Pathogens Institute, University of Florida, Gainesville, FL 32610, USA; kuttichantran@ufl.edu (K.S.); wisely@ufl.edu (S.M.W.); jgmorris@epi.ufl.edu (J.G.M.J.); 3Department of Infectious Diseases and Immunology, College of Veterinary Medicine, University of Florida, Gainesville, FL 32610, USA; 4Department of Wildlife Ecology and Conservation, University of Florida, Gainesville, FL 32611, USA; 5Department of Internal Medicine, Division of Infectious Diseases and Global Medicine, University of Florida Health/Shands Hospital, Gainesville, FL 32610, USA

**Keywords:** *Jeilongvirus*, viral emergence, paramyxovirus, rodent virus, cell culture

## Abstract

A novel jeilongvirus was identified through next-generation sequencing in cell cultures inoculated with spleen and kidney extracts. The spleen and kidney were obtained from a *Peromyscus gossypinus* rodent (cotton mouse) found dead in the city of Gainesville, in North-Central Florida, USA. Jeilongviruses are paramyxoviruses of the subfamily *Orthoparamyxovirinae* that have been found in bats, cats, and rodents. We designated the virus we discovered as Gainesville rodent jeilong virus 1 (GRJV1). Preliminary results indicate that GRJV1 can complete its life cycle in various human, non-human primate, and rodent cell lines, suggesting that the virus has a generalist nature with the potential for a spillover event. The early detection of endemic viruses circulating within hosts in North-Central Florida can significantly enhance surveillance efforts, thereby bolstering our ability to monitor and respond to potential outbreaks effectively.

## 1. Introduction

*Jeilongvirus* is a genus in the family *Paramyxoviridae* and the subfamily *Orthoparamyxovirinae* [1]. The family *Paramyxoviridae* comprises enveloped negative-sense single-stranded RNA viruses that infect birds, fish, mammals, and reptiles. Several paramyxoviruses affect humans, including measles, mumps, and parainfluenza viruses. Paramyxoviruses are frequently associated with respiratory infections and interspecies transmissions [2,3]. This attribute in the latter makes it important to identify and characterize these viruses, not only because of their potential cross-species transmission but also because some cause serious illnesses in humans, as exemplified by Hendra and Nipah viruses, zoonotic pathogens of concern [4]. In addition, surveillance of viruses circulating in wildlife populations is also crucial to understanding the evolutionary and ecological context of potentially zoonotic viruses before they jump into humans [5].

The genus *Jeilongvirus* was previously proposed to describe Beilong and J-viruses, whose genome structures differed moderately from those of existing genera within the *Orthoparamxovirinae* subfamily [1]. Jeilongvirus genomes differ from those of other paramyxoviruses in that they contain two additional transcription units, a gene for a small hydrophobic protein (*SH*) and a gene for a transmembrane protein (*tM*). The *SH* and *tM* genes are located between the fusion (*F*) and receptor-binding protein (*RBP*) genes, except in Mount Mabu Lopuromuys virus 1 and 2 which encode only the *tM* gene [6,7]. Transcription is highly regulated within *Paramyxoviridae*, which have genomes containing six to ten genes that encode the viral proteins [8]. Their genes are separated by intergenic regions with conserved non-coding sequences with termination, polyadenylation, initiation signals, and a trinucleotide motif (CUU in most jeilongvirus genomes) between the termination and initiation sites [9]. Although jeilongviruses were previously considered rodent viruses, they have recently been found in bats and felines, indicating that these viruses have a broader host range [9]. We describe the isolation of a novel jeilongvirus, its coding-complete sequence (CDS), and a preliminary host cell tropism study performed in various cell lines. The virus was isolated from a cotton mouse (*Peromyscus gossypinus*) found dead in Gainesville, Florida. Cotton mice are a rodent species in the family Cricetidae that is found in the woodlands in the southern USA. Analyses of the CDS of the virus we isolated provide support for a new species in the genus *Jeilongvirus*. We have named the new virus Gainesville rodent jeilong virus 1 (GRJV1). Virus replication studies performed in various cell lines indicate GRJV1 has a broad host species tropism.

## 2. Materials and Methods

### 2.1. Virus Source

On 2 May 2021, a domestic cat brought a deceased *Peromyscus gossypinus* mouse into the home of one of the authors in the city of Gainesville, which is in North-Central Florida, USA. The dead mouse was immediately transported to a laboratory for gross necropsy, and the kidneys, large intestines, liver, lungs, and spleen were frozen at −80 °C for future analyses. Ours was an opportunistic study, not a planned one, and the purpose of the collection of the mouse organs was to determine whether they might contain mule deerpox virus, as we would like to know whether rodents are vectors of that virus [10]. The domestic cat exhibited no clinical signs of disease and, therefore, was not studied in connection with this study.

### 2.2. Preliminary Virus Isolation Attempt in Vero E6 Cells

As we were interested in isolating mule deerpox, Vero E6 cells (*Cercopithecus aethiops* [African green monkey kidney] obtained from the American Type Culture Collection (ATCC, Manassas, VA, USA, cat. no. ATCC CRL1586) were chosen for primary isolation. After thawing, approximately 100 mg of each organ was homogenized in Advanced Dulbecco’s Modified Eagle’s Medium (aDMEM, Invitrogen Corp., Thermo Fisher Scientific, Waltham, MA, USA), supplemented with 2 mM L-alanyl-L-glutamine (GlutaMAX™, Invitrogen Corp., Thermo Fisher Scientific, Waltham, MA, USA) and antibiotics (PSN; 50 μg/mL penicillin, 50 μg/mL streptomycin, 100 μg/mL neomycin [Invitrogen Corp, Thermo Fisher Scientific, Waltham, MA, USA.]) to form 10% *w*/*v* suspensions using sterile manual tissue grinders (Fisher Scientific, Waltham, MA, USA). The homogenates were briefly centrifuged to pellet particulate material, and the supernatant was equally aliquoted into two tubes. One aliquot of each was subsequently filtered through a 0.45 µm pore size syringe tip filter (Grainger, Lake Forest, IL, USA) to remove contaminating bacteria and fungi. Thereafter, paired filtered and unfiltered homogenates were each inoculated at 200 µL onto nearly confluent monolayers of Vero E6 cells in T75 flasks (75 cm^2^ cell culture flasks, Corning Inc., Corning, NY, USA) containing cell culture medium that consisted of aDMEM supplemented with 10% low-antibody, heat-inactivated, gamma-irradiated fetal bovine serum (FBS, Hyclone, GE Healthcare Life Sciences, Pittsburgh, PA, USA), GlutaMAX, and PSN. The Vero E6 cells were obtained for other studies and validated as free of adventitious agents. The inoculated cells were incubated at 37 °C in a 5% CO_2_ environment and observed at 400× using an inverted microscope equipped with phase optics for an observation period of 30 days, with refeeds performed every three days, before being considered negative for virus isolation. If virus-induced cytopathic effects (CPEs) were observed, the spent growth media were collected, mixed with sterile filtered trehalose (used as a cryopreservation agent for virions) to a final concentration of 10% (*w*/*v*), and stored at −80 °C for future analyses.

### 2.3. Next-Generation Sequencing

After thawing frozen material collected from Vero E6 cells inoculated with spleen homogenate, RNA was extracted from virions therein using a QIAamp Viral RNA Mini Kit (Qiagen, Valencia, CA, USA) according to the manufacturer’s protocol. A cDNA library was generated using a NEBNext Ultra RNA Library Prep kit (New England Biolabs, Ipswich, Massachusetts, USA) and sequenced on an Illumina NextSeq 1000 sequencer. A de novo assembly of the paired-end reads was performed using MEGAHIT v1.1.4 [11]. The assembled contigs were then subjected to BLASTX searches against the National Center for Biotechnology Information’s (NCBI) non-redundant protein database using OmicsBox v1.2 (BioBam). Open reading frames within the assembled GRJV1 genome were predicted using ORFfinder [12]. The GRJV1 genome was subsequently annotated manually after comparison with those of other complete jeilongvirus genomes in GenBank, with attention paid to proteins arising from co-transcriptional editing [13,14,15] as relevant to jeilongviruses [16].

### 2.4. Phylogenetic and Genetic Analyses

The *large* (“*L*”) gene in members of the family *Paramyxoviridae* encodes for the RNA-dependent RNA polymerase (L protein), whose amino acid sequence is highly conserved, and its nucleotide sequence is used to distinguish between species within the family [4]. Amino acid sequences of the deduced L proteins of 47 jeilongviruses and bank vole virus 1 (as an outgroup) were retrieved from the NCBI GenBank database. They were then aligned with GRJV1 using MAFFT aligner within Geneious Prime v2022.2.2 [17]. The Maximum Likelihood phylogenetic trees were constructed in MEGA 11 with 1000 bootstraps performed to test the robustness of the clades [18]. A pairwise genetic distance analysis was also performed on the *L* gene sequences of the 47 jeilongviruses using the Sequence Demarcation Tool version 1.2 with the MAFFT alignment option implemented [17].

### 2.5. Real-Time RT-PCR Assay

A real-time RT-PCR (rRT-PCR) assay that targets the *L* gene of GRJV1 was developed to gauge viral host species tropism by measuring virus yields in cell lines derived from various animal species. Virus RNA (vRNA) extracted from virions in the spent culture medium using a QIAamp Viral RNA Mini Kit was used for the assay, along with the primers and probe listed in Table 1. A SuperScript™ III One-Step RT-PCR system with Platinum™ *Taq* DNA Polymerase (Thermo Fisher Scientific, Waltham, MA, USA)) was used for rRT-PCR. Briefly, 6 µL of RNA–primer mix (5 µL of the purified vRNA and 1 µL of primer mix containing 5 pmol each of forward and reverse primers) was denatured for 5 min at 67 °C and then rapidly cooled to 4 °C. Thereafter, dNTPs, reaction buffer, and enzyme mix were added, and rRT-PCR was performed in a total volume of 25 µL. The rRT-PCR assays were performed in a BioRad CFX96 Touch Real-Time PCR Detection System. Cycling conditions were as follows: cDNA synthesis for 15 min at 50 °C, 2 min at 95 °C to activate Taq polymerase, followed by 40 cycles of denaturation for 15 s at 95 °C, annealing at 54 °C for 20 s and 25 s at 72 °C for extension, with a final extension step of 72 °C for 5 min, then 4 °C for ∞. An example of rRT-PCR amplification curves resulting from use of the primers we designed is presented in Appendix A.

### 2.6. Host Species Tropism Studies

For this part of the experiment, 50 µL aliquots, each containing 5 × 10^3^ PFU of GRJV1, were inoculated onto newly confluent monolayers of A549 (*Homo sapiens* [Adenocarcinomic alveolar] cells, ATCC CCL cat. no. 185), BHK-21 (*Mesocricetus auratus* [Hamster] kidney cells, CCL cat. no. 10), Caco-2 (*Homo sapiens* [Adenocarcinomic colorectal] cells, HTB cat. no. 37), Calu-3 (*Homo sapiens*, lung cells HTB Cat. no. 55), CV-1 (*Cercopithecus aethiops* [African green monkey] kidney cells CCL cat. no. 70), HeLa (*Homo sapiens* cervix cells, CCL cat. no. 2), LLC-MK2 (*Macaca mulatta* [Rhesus monkey] kidney cells, CCL cat. no. 7), MDCK (*Canis familiaris* [Dog] kidney cells, CCL cat. no. 34), MRC-5 (*Homo sapiens* lung cells, CCL cat. no. 171), Neuro-2a (*Mus musculus* [Mouse] brain cells, CCL cat. no. 131), OHH1.K (*Odocoileus hemoinus hemionus* [Columbian black tail deer] kidney cells, CRL cat. no. 6193), SAEC (*Homo sapiens* lung cells, PCS cat. no. 301-010), and Tb1Lu (*Tadarida brasiliensis* [Free-tailed bat] lung cells, CCL cat. no. 88) cultured in 3 mL of complete cell growth medium consisting of aDMEM supplemented with 10% heat-inactivated, gamma-irradiated fetal bovine serum (FBS) (GE Healthcare Life Sciences, Pittsburgh, PA, USA, cat. no. SH30070.03IH), GlutaMAX, and PSN. The inoculated cells were subsequently incubated at 37 °C in a humidified 5% CO_2_ environment. Cells were refed with reduced FBS media (3% FBS and the same other components) every 3 days post-inoculation (dpi). Once virus-induced CPEs were identified or at 6 dpi (whichever came first), the spent cell growth media were sampled and stored at −80 °C for virus quantification assays and rRT-PCR tests.

### 2.7. Plaque Assays

Virion yields at 6 dpi for each cell line inoculated with GRJV1 were determined by plaque assays using CV-1 cells as the indicator cell line. CV-1 cells were chosen as indicator cells because the virus-induced CPEs in these cells were more apparent than were observed in the other cell lines. The CV-1 cells were seeded into 12-well plates (GenClone, Genesee Scientific, Morrisville, NC, USA), after which the cells were placed in a humidified 37 °C incubator with 5% CO_2_ overnight (Appendix B). At confluency, the cell growth medium was removed and 200 µL of serially diluted spent cell medium was applied onto the CV-1 monolayers. Subsequently, the plates were placed in a humidified 37 °C incubator with 5% CO_2_ and rocked every 15 min for 1 h. Following 1 h, 1 mL of Avicel-supplemented media (Appendix B) was added to each well. The 12-well plates were subsequently incubated in a humidified 37 °C incubator with 5% CO_2_ for 5 days. After 5 days, the wells were each washed with 0.5 mL of PBS (Gibco, pH 7.4, ThermoFisher Scientific, Waltham, MA, USA) to remove the Avicel overlays before 0.5 mL of crystal violet fixative (49% Acros Organics Methyl Alcohol [ThermoFisher Scientific, Waltham, MA, USA], 49% Optima Acetone [ThermoFisher Scientific, Waltham, MA, USA], and 1% (*w*/*v*) crystal violet [Sigma Aldrich, St. Louis, MO, USA]) was applied for 1 h. This was performed in triplicate per cell type.

## 3. Results

### 3.1. Initial Isolation of Virus in Vero E6 Cells

Mixed CPEs including cell death and the formation of syncytia were observed by 6 dpi of Vero E6 cells inoculated with kidney and spleen tissue homogenates but not in cells inoculated with homogenates of other organs or in mock-infected cells maintained in parallel).

### 3.2. Sequencing Results

Next-generation sequencing was performed three times on the virus genome obtained from the spleen culture. Our most successful and final attempt produced 7,868,176 reads, and de novo assembly using MEGAHIT v1.1.4 resulted in a 17,021 ribonucleotide (rnt) GRJV1 genome. This indicated that perhaps one ribonucleotide at either end of the genome was not determined, since paramyxovirus genomes typically exist with ribonucleotide lengths that are multiples of six [16]. We performed 3′ and 5′ RACE reactions to verify the untranslated regions at the 3′ and 5′ ends and found that one ribonucleotide at the 3′ end of the negative genome varied (A, C, G, or U). Due to uncertainty, we have deposited only the 17,021 rnt sequence in GenBank, under accession number PP883550. A BLAST search indicated that the virus genome that had been sequenced had a 74.37% identity with the coding-complete sequence (CDS) of Wenzhou Apodemus agrarius jeilong virus 1. Ten GRJV1 genes were identified by in silico analyses and by comparisons with annotated jeilongvirus genomes deposited in GenBank, and the deduced arrangement of the coding regions is depicted in Figure 1. Some of GRJV1’s important genomic features and their nucleotide (nt) positions, as well as basic characteristics of its deduced proteins, are given in Table 2.

The deduced V protein of GRJV1, which is coded by viral mRNA wherein one G residue is added at the P/V/C editing site of the P gene coding sequence (Table 2) [16], has seven cysteine residues (“cys cluster”) at the carboxy terminus (Figure 2A). The cys cluster is a conserved feature of various paramyxovirus V proteins, and the GRJV1 cys cluster aligns with those of Beilong virus (Figure 2B) and other paramyxoviruses, such as Ninove microtus virus, Pohorje myodes paramyxovirus 1, Meliandou mastomys virus, and Jingmen Apodemus agrarius jeilong virus 2.

### 3.3. Phylogenetic Analyses

A Maximum Likelihood (ML) analysis based on the L protein amino acid sequence generated a well-supported tree, wherein GRJV1 formed a unique branch within the genus *Jeilongvirus* (Figure 3). The amino acid identity of GRJV1 L protein ranged from 49.7 to 69.6% compared to those of other jeilongviruses, and the highest amino acid identity was observed with the L protein of Meliandou mastomys virus (Appendix A; GenBank accession no. OK623361).

### 3.4. Host Cell Tropism and Virus Yields

Virus-induced CPEs were identified in most of the cell lines by 6 dpi. Cytopathic effects included the formation of syncytia and cytoplasmic vacuoles followed by cell death (Figure 4). In some cell lines, such as CV-1 and MRC-5, elongation of the cells was pronounced (Figure 4D). Of all the cell lines inoculated with GRJV1, CPEs were most apparent in human, non-human primate, and rodent cell lines.

Plaque assays revealed that at least 10^2^ PFU/mL was obtained in each of the cell lines that were tested, indicating that cells from a broad range of host species are susceptible and permissive to this virus and that the highest quantity of virions was produced in CV-1 and LLC-MK2 cells (Table 3). An example of the results of plaque assays are shown in Appendix C. As shown, relatively small plaques were observed under the conditions that were used for the assay.

### 3.5. rRT-PCR Assay

Spent media from cells inoculated with GRJV1 tested positive by rRT-PCR, and non-inoculated cells maintained in parallel tested negative (Appendix A). Furthermore, the results of the rRT-PCR tests coincided with those of the viral quantification assay, with the exception of Tb1Lu cells. The rRT-PCR assay indicated that the highest levels of replicated GRJV-1 genomic RNA were detected in human, non-human primate, and rodent cell lines.

## 4. Discussion

We report the discovery of GRJV1, a virus that has a generalist nature, able to replicate in cells from different species. The GRJV1 genomic sequence we determined consists of at least 17,021 ribonucleotides that were reproducibly determined by NGS. The genome sequence we determined may lack one ribonucleotide, which we were unable to unambiguously identify through 3′ and 5′ RACE reactions. Only paramyxovirus genomes of integer hexamer length replicate efficiently and are found naturally, indicating that the GRJV1 genomic sequence we determined is incomplete, possibly lacking only one ribonucleotide to attain integer hexamer length, though others may also be missing. Therefore, we designated the sequence we unambiguously identified as a coding-complete sequence (CDS), not a complete genomic sequence.

Current information on genus *Jeilongvirus* indicates that natural infection is restricted to small mammals (bats, cats, rodents). A recent study in China, which aimed to understand the host distribution of jeilongviruses in wild rodents, determined that the distribution of the viruses was widespread in diverse regions and rodent species [19]. However, the host range for each type of jeilongvirus is likely broader. For example, antibodies for J paramyxovirus were detected in wild mice, rats, pigs, and human populations in Australia [20]. Our in vitro studies suggest that GRJV1 has a broad host range tropism; as for J paramyxovirus, further studies are necessary to explore the extent to which GRJV1 also affects other animals. Continued surveillance of these viruses is warranted to establish whether the virus affects other animals, especially as it is not known whether jeilongviruses cause illnesses in their hosts, and the virus may be zoonotic. Animal studies are also needed to explore whether GRJV1 induces illnesses in its resumed natural host and other animals.

Information is still lacking on GRJV1’s transmission patterns, cell surface receptors, and cell and organ tropism within the infected host. These could be topics for future research. Some jeilongviruses recently identified in North American wildlife may have sialic acid (SA) virus receptors. The SAs are linked to glycoproteins and gangliosides, serving as receptors for several human and animal viruses [5,21]. Sialic acids have conserved biochemical structures and are widespread across all vertebrate hosts [21]. Such receptors are used by RNA viruses such as corona-, flavi-, paramyxo-, picorna- and reoviruses, as well as the influenza virus, to enter into host cells [21]. Sialic acid receptors are used by various zoonotic viruses, such as the highly pathogenic H5N1 avian influenza virus, and the redundancy in receptors between animal species provides an opportunity for viruses to jump species and adapt to the human host [21,22]. Efforts should be made to characterize the cell surface receptors used by jeilongviruses to better estimate the potential host range of these viruses. It is of interest that the relative amount of virus RNA in TB1Lu cells is high compared to the quantity of progeny virions that are released into the spent media. This may indicate inefficient release of the virions from infected cells or some block to efficient packaging of GRJV1 RNA into virus particles. Such an observation may provide insight as to why jeilongviruses might be carried by bats without inducing clinically apparent health consequences. More work needs to be performed to explain this finding and to determine whether this also occurs in other bat cells or is specific to the Tb1Lu cells.

*Paramyxoviruses* should be considered of high concern as potential spillover pathogens, as various species within the *Paramyxoviridae* family demonstrate the ability to establish infections in humans, and these viruses exhibit remarkable receptor tropism flexibility through the course of their evolution [4]. Plaque assays indicated that GRJV1 virion yields were highest in kidney and lung cell types from human, non-human primate, and rodent cell lines. The CPEs produced by this virus varied depending on the cell type, but cell killing was evident in most cell lines. The results of our in vitro tests suggest that GRJV1 has broad cellular and host species tropisms. More experiments must be performed to determine if this virus can establish infections outside of a rodent host. The virus, for example, might be a rare cause of respiratory infections in humans that come into contact with rodent excreta, such as the case with hantaviruses and residents of the American Southwest.

## Figures and Tables

**Figure 1 pathogens-13-00831-f001:**
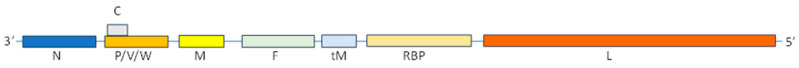
Genome organization (3′-to-5′) of GRJV1. Each box represents a separate coding sequence and is drawn to approximate scale. Slashes (/) indicate where multiple distinct ORFs are present within mRNA transcripts. Co-transcriptional editing leads to expression of the P, V, or W proteins. N, nucleocapsid protein; P, phosphoprotein; V, multifunctional and interferon antagonist protein; W, protein that acts as a neutralizer of host interferon response; C, regulates viral transcription, replication, and other functions; M, membrane protein; F, fusion protein; tM, transmembrane protein; RBP, receptor-binding protein; L, virus polymerase protein (RdRp, RNA dependent RNA polymerase). The V and W proteins arise from RNA editing of the P gene, whereas the C protein is a product of leaky scanning of the P gene.

**Figure 2 pathogens-13-00831-f002:**
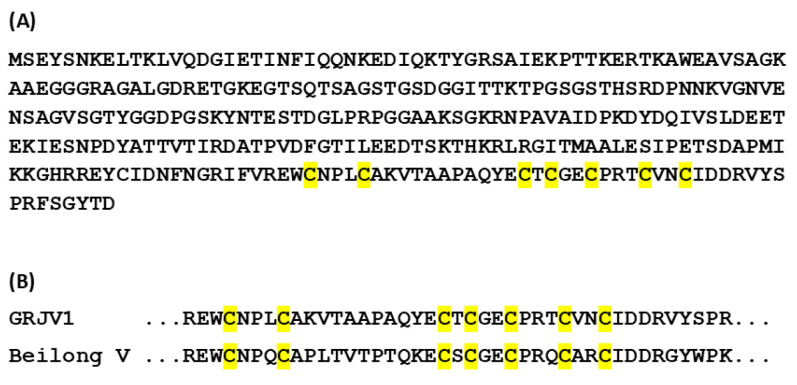
Basic features of the deduced GRJV1 V protein. (**A**) V protein amino acid sequence. The cysteine residues that comprise the cys cluster are highlighted in yellow. (**B**) Alignment of the cys cluster of GRJV1 with that of Beilong virus.

**Figure 3 pathogens-13-00831-f003:**
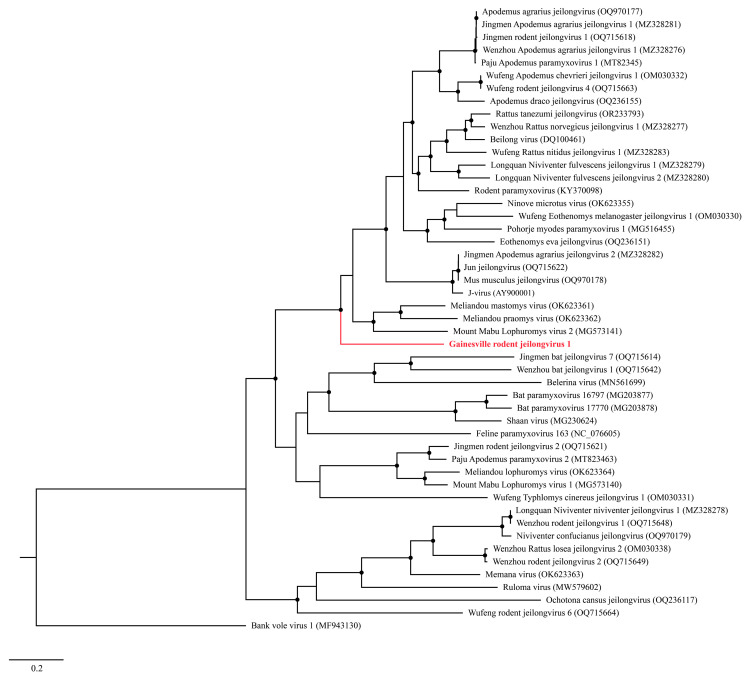
Maximum Likelihood phylogram depicting the relationship between the Gainesville Peromyscus gossypinus jeilongvirus (highlighted in red) and 47 other jeilongviruses and bank vole virus 1 (as an outgroup), based on the amino acid sequence alignments of the L protein. Nodes with black circles are supported by bootstrap values ≥ 80%.

**Figure 4 pathogens-13-00831-f004:**
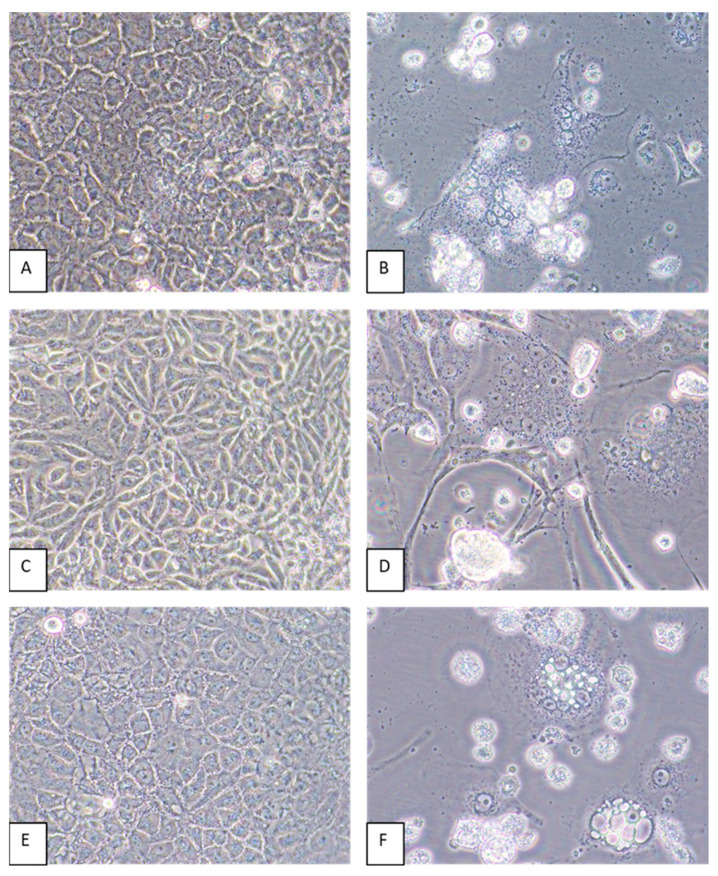
A549, CV-1, and LLC-MK2 cells inoculated with GRJV1. (**A**) Mock-infected A549 cells, 6 days post-inoculation (dpi). (**B**) Appearance of A549 cells inoculated with GRJV1 6 dpi. Dead cells, perinuclear darkening, and syncytia are evident. (**C**) Mock-infected CV-1 cells, 6 dpi. (**D**) CV-1 cells inoculated with GRJV1 6 dpi. Cell elongation, cytoplasmic vacuoles, dead cells, perinuclear darkening, and syncytia are evident. (**E**) Mock-infected LLC-MK2 cells, 6 dpi. (**F**) LLC-MK2 cells inoculated with GRJV1 6 dpi. Dead cells, intracellular vacuoles, perinuclear darkening, and syncytia are evident. Original images were taken at 400× magnification.

**Table 1 pathogens-13-00831-t001:** Sequences of the primers and probe used in this study.

Oligonucleotide ID	Sequence (5′ to 3′)
RodparamyxF	CCT CCA TGG TTT GGA ACA CAT GCC
RodparamyxP	6-FAM-CCT GGG GGA GGG ATC AGG GGC AAT GC-BHQ-1
RodparamyxR	GCC TCA GAA GGG TAG ACA TGG TG

6-FAM, 6-carboxyfluorescein; BHQ-1, Black Hole Quencher-1.

**Table 2 pathogens-13-00831-t002:** Genomic features and characteristics of the deduced proteins of GRJV1 (GenBank no. PP883550).

Gene	Length (Ribonucleotides)	Genomic Positions	Gene Product	Amino Acid Length
Leader sequence	93	1–3	N/A ^a^	N/A
N	1647	31–1677	Nucleocapsid protein	548
TIS ^b^	3	1774–1776	N/A	N/A
P	1470	1863–3332	Phosphoprotein	489
P/V/C editing site	16	2551–2566	N/A	N/A
V	894 ^c^	1863–2755	V protein	298
C	456	1906–2364	C protein	152
W	924 ^d^	1863–2784	W protein	308
TIS	3	3509–3511	N/A	N/A
M	1017	3550–4566	Matrix protein	339
TIS	3	4829–4831	N/A	N/A
F	1641	4975–6615	Fusion protein	546
TIS	3	6631–6633	N/A	N/A
tM	777	6745–7521	Transmembrane protein	258
TIS	3	7701–7703	N/A	N/A
RBP	2352	7753–10,104	Receptor-binding protein	783
TIS	3	10,290–10,292	N/A	N/A
L; RdRp	6759	10,363–16,941	Large; RNA-dependent RNA polymerase	2192
Trailer sequence	530	16,492–17,021	N/A	N/A

^a^ N/A; not applicable. ^b^ TIS; trinucleotide intergenic sequence. ^c^ V; RNA editing adds an extra G to the editing site. ^d^ W; RNA editing adds two Gs to the editing site.

**Table 3 pathogens-13-00831-t003:** Viral quantification results.

Cell Line	Virus Titer, Log_10_PFU/mL *
A549	5.3 ± 0.3
BHK-21	5.9 ± 0.4
Caco-2	5.3 ± 0.7
CV-1	6.8 ± 0.1
HeLa	4.6 ± 0.7
LLC-MK2	7.1 ± 0.1
MDCK	4.0 ± 0.3
MRC-5	6.4 ± 0.7
Neuro-2a	3.5 ± 1.5
OHH1k	2.9 ± 0.1
SAEC	6.0 ± 0.6
Vero E6	7.2 ± 0.3
Tb1Lu	2.6 ± 0.4

* Average of three replicates.

## Data Availability

The original contributions presented in the study are included in the article/Appendix A. Further inquiries can be directed to the corresponding author/s.

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
