# Peer review of "A Novel Jeilongvirus from Florida, USA, Has a Broad Host Cell Tropism Including Human and Non-Human Primate Cells"

_pathogens, 2024, doi:10.3390/pathogens13100831_

Round 1

Reviewer 1 Report

Comments and Suggestions for Authors

In their manuscript “A Novel Jeilongvirus from Florida, USA, has a Broad Host-Cell Tropism Including Human and Non-Human Primate Cells”, DeRuyter et al., present the isolation of a new jeilongvirus, sequencing data of the virus isolate and first studies in cells to determine the host cell tropism.

The presentation and characterization of new virus isolates is of importance to also prepare for the emergence of new zoonotic viruses. However, the (amount of) data presented as well as the presentation itself lack of detail and care and need major revision.

Points to address:

Minor: There are a few sentences that make no sense or should be rephrased for better understanding. The manuscript should be carefully edited with regards to that. For example, the first sentence of the abstract is hard to understand, here 2 sentences would be better (ll. 14-17). Semicolon in l. 48?, l. 156 was identified. L. 161 CPE was more apparent. L.209 CPE was identified. L.235 spin-dling?    ...

Paramyxoviruses are not generally known for “frequent inter-species transmission” (l. 30). The family includes strict animal pathogens, human pathogens and also zoonotic pathogens. E.g. it is a hallmark for mumps or measles to only infect humans. Please rephrase.

The last paragraph is a wild mixture of information belonging to the introduction, results and repeats, and already summarizes all relevant findings. Please structure and rephrase.

The explanation in lines 63-66 sound very odd. This should be shortened.

I am not familiar with this mouse species, but 100 mg of organ material sounds like a huge amount?

Why was the whole genome sequencing only performed with material originating from spleen?

The whole genome sequence should be completed and then uploaded to Genbank for reference by others.

I was not able to retrieve and look at any of the mentioned Supplementary Figures. However, I am not sure how relevant a PCR amplification curve is for the readership.

The relevance of the Appendix A (why A when there is no B) is not clear to me. It is with the exception of cell density more or less the same as the method described in the method section. Include this detail and remove the Appendix.

l.174 “this experiment”. The titration was performed in triplicates? This is not an experiment, but part of an experiment.

In the “virus isolation” section, authors describe that cells are observed for 30 days to monitor CPE. In the results section, authors write that CPE was observed at 6 DPI. What is correct?

One major point is that authors do not show any picture of the first virus isolation. This is one of the major findings of the paper and data should be shown!

Figure 1: It would be nice if the length of the respective open reading frames is indicated/written down somewhere. What is the length of the untranslated regions (5’ UTR, 3`UTR). It is not explained how authors conclude that the P gene also encodes for C/V/W? Did they check for a second open reading frame? Are there editing sites? Are there multiple mRNAs produced by the P gene that have been detected?

Description of a result as “CPE were identified in most cell lines” and then refer to a Figure, in which only 3 cell lines are shown, is very superficial and not scientific writing. Please be precise in description of all results. And CPE stands for cytopathic effect and is thus singular.

This is to be continued with the description of the data from the titrations. Looking at Table 2, I am not sure if CV-1 and LLC-MK2 cells are the cells with the highest number of virus produced. Titer in Vero E6 cells? In addition, authors describe that cells have been inoculated with 5x10^3 PFU. However, they don’t mention removal of inoculum and washing of cells. Thus, I wonder, if the cells with low titers in the supernatant really produced new virus or if this is leftover inoculum. Please clarify. To distinguish this, one could perform growth kinetics and not just determine one single timepoint. The overall conclusion (ll.265/266) are therefore questionable.

Figure 3: Which Figure shows which cells? A549 Mock are not shown? LLC-MK2 cells Mock twice? Please correct.

Why are the genome ends not sequenced? Not even attempted to obtain? There is a virus isolate and thus, enough material to attempt sequencing of the genome ends? Also, if the sequence obtained not follows the rule of six, attempts should be made to obtain a full and correct sequence that can then be deposited into Genbank. This should be done.

ll. 258 and following: please be more specific in discussing paramyxoviruses and spillover events. In general, paramyxoviruses are considered to be species-specific with few exempts.

ll. 262 and following: in contrast to results, authors here mention also rodent cell lines to produce high amounts of virus. Is the results section or the discussion section correct?

Overall, the discussion is a repeat of the results (or the missing results) and barely a structured discussion.

Comments on the Quality of English Language

Sentences should be rephrased. Singular versus plural correction in verbs.

Reviewer 2 Report

Comments and Suggestions for Authors

The study by DeRuyter et al. focuses on the isolation and genetic characterization of a new member of the Paramyxoviridae family, discovered by chance in a dead rodent. The work is well-written and meets most of the standard criteria for a study of this type. However, I have a few comments:

1.        I suggest adding the specific geographic location of the discovery to the methodology. While the title mentions Florida, the exact location where the rodent was found is not clearly stated.

2.        It would be helpful to clarify whether the cat involved exhibited any clinical symptoms or if it was also studied in connection with the discovery.

3.        I recommend including the accession number of the sequence in the GenBank database.

4.        In Figure 3A, are these A549 or LLC-MK2 cells? The figure caption seems to contain an error. It would be useful to specify the type of microscopy used in the figure caption.

5.        I consider it important to mention some characteristics of the lytic plaques formed by the new virus, and if possible, to include some photographs.

6.        I also recommend including members of the main genera of the subfamily in the phylogenetic tree to better illustrate the evolutionary distances and relationships.

7.        It's important to discuss some limitations of the study. For example, the need to verify the sequence ends to confirm the rule of six is mentioned, but I also suggest noting the lack of electron microscopy to visualize the viral particles as a limitation.

8.        In section 3.3, the term "ML analysis" is mentioned without explanation. Please clarify what ML means.

9.        The study refers to supplementary material, including RT-PCR curves and an identity matrix of the L protein compared to other jeilongviruses. I could not locate these figures, so they are not included in the current evaluation of this article. However, I believe the second figure would benefit from including sequences from other genera, as suggested for the phylogenetic tree.

Reviewer 3 Report

Comments and Suggestions for Authors

Thank you very much for the opportunitity to revise the original manuscript entitled  A Novel Jeilongvirus from Florida, USA, has a Broad Host-Cell Tropism Including Human and Non-Human Primate Cells.

The research presents evidence for a new Jeilongvirus, isolated from cotton mouse in Florida with a comprehensive NG sequencing, phylogenesis and various human and non-human cell lines susceptibilty. The spillover potential is highlighed.Having the intersting and attractive story of the sample origin that study appears to be a serendipity, which is great. The paper is written in an excellent style and language, clearly formulated title and well-described methods and results, as well as a short discussion which is sufficient, giving the limited data on virus transmission details. References are relevant.

Here are my comments:

1.The introduction section could be expanded a little to provide more literature data on virus biology, host patterns, replication cycle and transmission patterns, pathology induced and if available, previous studies on the isolation and infection  in vitro and in vivo.

2. Figure 2. representing the dendrogram of GRJV1 has poor resolution and should be improved - it is not legible, even if zoomed in.

3. References are adequate and relevant but are too limited in number, i.e.14,it appears to me.

4. Giving the broad-spectrum of susceptible cells it would be interesting to check if the infectivity of the virus will be preserved by serial passages in these line and if it increases or decreases. Sometimes viruses lose their fitness by multiple re-infections being unable to adapt to the host. That is just as a suggestion for possible further research. 3D cells or organoids, as well as in vivo studies would be also crucial for the characterization of GRJV1.

Round 2

Reviewer 1 Report

Comments and Suggestions for Authors

In the revised version of A Novel Jeilongvirus from Florida, USA, has a Broad Host-Cell Tropism Including Human and Non-Human Primate Cells”, DeRuyter et al., addressed several points raised in the first round of reviews. However, some important points remain open or are not sufficiently addresses:

Comment 11: If authors insist on keeping their amplification “curves” in the appendix, they should be well aware of the fact, that these “curves” do not exhibit curves that should be expected using RT-qPCR since they are flat and not the typical shape. This demonstrates that the PCR is not well established and most likely, too much primer/probes are used. So overall, this appendix weakens the paper.

Comment 12: Sure, appendix A can be kept, but the only really relevant information is cell density which could easily be included into material and methods. Which supernatants are tested in the plaque assay is not a method, this should become clear in the manuscript. And this is still a point lacking.

Comment 16: Please include the fact that the analysis is done by in silico analyses. And why does the reader need to go to Genbank to retrieve information that could clearly be included in this manuscript? This is important information! And no, it is virus genome information, that does not vary between cells in a host.

Comment 17: The reviewer is well aware of the fact that CPE can differ in different cell lines and surprisingly, also by different viruses. However, authors don’t not describe their results appropriately.

The relevance of any of the additional information is questionable. What should the reader get out of e.g. Appendix B? No explanation, no labeling…?

Overall, results remain poorly described and lack important information. The discussion is still weak.

Comments on the Quality of English Language

Minor editing required, mostly spelling mistakes.

Reviewer 2 Report

Comments and Suggestions for Authors

I have reviewed the updated version and can confirm that the authors have made the suggested changes or addressed the comments. I have no further comments or questions.

Author Response

No comments made by reviewer.